# High-veracity functional imaging in scanning probe microscopy via Graph-Bootstrapping

Xin Li[1,2,3,4], Liam Collins[1,2], Keisuke Miyazawa[5], Takeshi Fukuma[6], Stephen Jesse[1,2] & Sergei V. Kalinin [1,2]

The key objective of scanning probe microscopy (SPM) techniques is the optimal representation of the nanoscale surface structure and functionality inferred from the dynamics of the cantilever. This is particularly pertinent today, as the SPM community has seen a rapidly growing trend towards simultaneous capture of multiple imaging channels and complex modes of operation involving high-dimensional information-rich datasets, bringing forward the challenges of visualization and analysis, particularly for cases where the underlying dynamic model is poorly understood. To meet this challenge, we present a data-driven approach, Graph-Bootstrapping, based on low-dimensional manifold learning of the full SPM spectra and demonstrate its successes for high-veracity mechanical mapping on a mixed polymer thin film and resolving irregular hydration structure of calcite at atomic resolution. Using the proposed methodology, we can efficiently reveal and hierarchically represent salient material features with rich local details, further enabling denoising, classification, and high-resolution functional imaging.

[1] Institute for Functional Imaging of Materials, Oak Ridge National Laboratory, Oak Ridge, TN 37831, USA. [2] Center for Nanophase Materials Sciences, Oak Ridge National Laboratory, Oak Ridge, TN 37831, USA. [3] Department of Industrial and Manufacturing Engineering, Florida State University, Tallahassee, FL 32306, USA. [4] Department of Statistics, Florida State University, Tallahassee, FL 32306, USA. [5] Division of Electric Engineering and Computer Science, Kanazawa University, Kakuma-machi, 920-1192 Kanazawa, Japan. [6] Nano Life Science Institute (WPI-NanoLSI), Kanazawa University, Kakuma-machi, 920-1192 Kanazawa, Japan. Correspondence and requests for materials should be addressed to S.V.K. (email: sergei2@ornl.gov)

After the first demonstration of atomic force microscopy (AFM) by Binnig and Rohrer[1], AFM and related force-based scanning probe microscopies (SPMs) have rapidly become the tool that opened the nanoworld for exploration and modification[2]. Since the late 1980s, a variety of SPM methods sensitive to magnetic[3], electrostatic[4–8], mechanical[9–11], and piezoelectric[12–15] properties of surfaces have been realized, followed by the development of a number of spectroscopic modes that provided insight into kinetics and thermodynamics of single-molecule reactions[16–19] and mechanisms of bias-induced phase transitions[20–24] on a single defect level.

Despite this progress, for over 20 years, the progress in SPM methods was associated preponderantly with the development of low noise and controlled environment platforms, as well as functionalized probes. At the same time, the basic principles of signal processing and visualization involved in SPM imaging remained the same, namely the use of the single frequency heterodyne detection methods in lock-in and phase-locked loop detection, and plotting-associated amplitude/phase or frequency maps. Considerable progress in SPM instrumentation was achieved in the early 2000s, with the introduction of dual frequency methods by Garcia[25] and Proksch[26]. The next step was the development of the band excitation (BE) method by Jesse and Kalinin[21, 27–31], which enabled quantitative measurements of conservative and dissipative interactions on the nanoscale. Since then, an increasing number of multifrequency SPM techniques including, amongst many others, bi-/tri-modal SPM[32, 33] and intermodulation techniques[34, 35] as well as multidimensional approaches such as three-dimensional force mapping AFM[36–38] (3D-AFM), holography[39], and time-resolved[40–42] techniques have been realized. Finally, the general acquisition (G)-Mode[43–45] approach has recently been developed to enable full capture of the information stream from the photodetector, potentially reaching the information limit of SPM.

However, multifrequency/multidimensional techniques such as intermodulation SPM[34, 35], BE[27–31], and G-Mode[43–45] generate large, often complex, datasets, necessitating approaches for visualizing and converting the data to materials specific information. In prior BE work, we have primarily used functional fitting of the data in the Fourier domain[27–31], relying on a prior based on simple harmonic oscillator physics (SHO model) of the tip–surface interactions. This approach by definition ignores the materials behaviors associated with deviations from SHO models, for example, nonlinearities that lead to dynamic stiffening or softening of the tip–surface junction and hence require more complex dynamic models. The introduction of such models, in turn, can lead to significant issues such as spurious growth in the number of free parameters, expansive analysis times, potential overfitting, etc. Meanwhile, the linear unmixing methods based on principal component analysis[46] show only limited usefulness, since the BE signal is non-linear with respect to the local mechanical properties.

Here, we propose and implement an approach based on low-dimensional embedding of high-dimensional data via a combination of graph analytics and hierarchical clustering, illustrated for BE-SPM and 3D-AFM but generally applicable to other data-rich SPM modalities. We note that fundamental physics of the tip–surface interactions is intrinsically low-dimensional and is determined by a relatively small number of materials parameters. The transfer function of the cantilever is a non-linear[11, 47] (and generally very complex) function of these parameters, precluding the use of the linear unmixing methods for the analysis. However, we argue that the intrinsic low dimensionality of the physics suggests the presence of the low-dimensional manifold can be derived from the high-dimensional response space of SPM measurements. Figure 1 illustrates the concept of manifold-physics inference applied to functional imaging using SPM.

## Results

**Traditional BE-SPM Imaging.** Unlike single frequency techniques, which excite/detect the tip–sample interaction within a single frequency bin, BE detection utilizes a non-sinusoidal excitation signal having finite amplitudes over a selected frequency space. Practically, this is achieved using a digitally

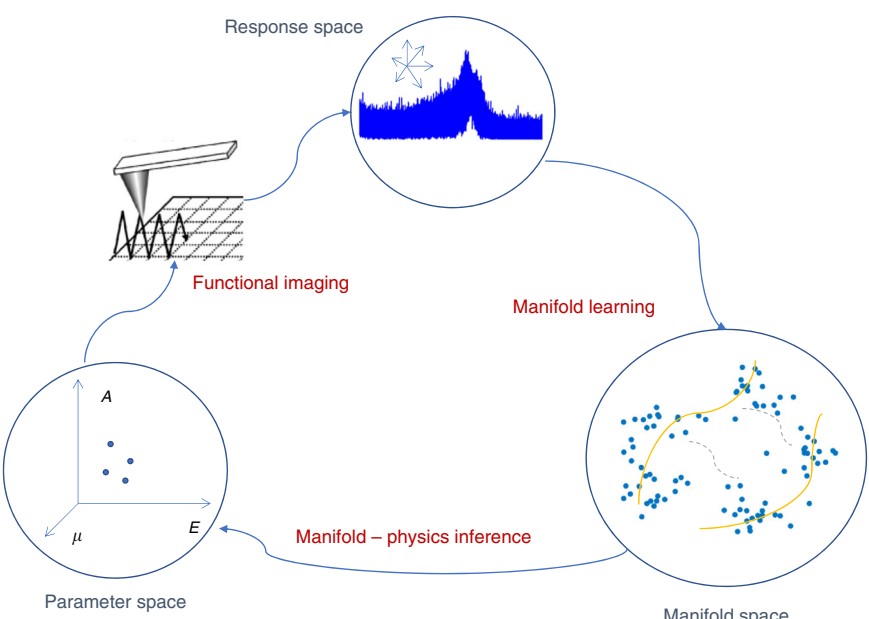

**Fig. 1** Concept of manifold-physics inference. The properties of the surface that define the scanning probe microscopy (SPM) signal form a low-dimensional parameter space, for example, defined by the Young and Poisson moduli and the work of adhesion. These properties are translated onto a (very high dimensional) response space by means of the imaging mechanism of the cantilever/microscope. Aside from possible discontinuities due to changes in imaging regimes, points that are in close proximity in the parameter space will generally be in close proximity in the response space, forming a complex non-linear manifold

synthesized signal having a defined band of frequencies (in the Fourier domain), which are subsequently inverse fast Fourier transformed (iFFT) to generate a signal in the time domain, used to modulate the tip–sample interaction. Further, the band of frequencies is typically chosen to be positioned (in the Fourier domain) across one or many contact resonance (CR) peaks, allowing further insight into the cantilever dynamics than accessible in single frequency techniques.

In the linear regime, the behavior of the cantilever CR can be approximated by a SHO and described by three independent parameters: CR frequency, $\omega_0$, amplitude at the CR, $A_0$, and quality factor, $Q$, which can be deconvoluted and stored as images as given by Eqs. (1) and (2):

$$A(\omega) = \frac{A^{\max} \omega_0^2}{\sqrt{(\omega^2 - \omega_0^2)^2 + (\omega\omega_0/Q)^2}} \quad (1)$$

$$\tan(\phi(\omega)) = \frac{\omega\omega_0/Q}{\omega^2 - \omega_0^2}. \quad (2)$$

In this way, knowledge of the full CR behavior in BE allows both conservative ($\Delta\omega_0$) and dissipative ($\Delta Q$) contributions of the tip–sample interaction to be decoupled. In terms of mechanical property measurements, this enables separating the influence of elastic (conservative) and viscous (dissipative) material behavior. This makes BE-SPM (sometimes referred to as CR-SPM) a promising route for mapping local mechanical properties of materials.

Figure 2 represents the topography and the SHO fitting results of the BE measurements on a thin-film polymer blend of polycaprolactone (PCL) in a polystyrene (PS) matrix. In Fig. 2b,

the amplitude response is seen to be reasonably constant across the entire image, with abrupt changes in the amplitude response detected at the interfaces between materials, likely due to sudden changes in sample topography leading to imperfect tip–sample contact. At the same time, clear evidence of differences in viscoelastic behavior of the polymer blend can be observed in the resonance frequency and $Q$ maps, Fig. 2c, d, respectively. The PS matrix clearly shows a higher resonance frequency and $Q$ factor than observed in the PCL inclusions. The shift in the CR to a higher frequency indicates an increase in the stiffness (Young's modulus) of the material. Whereas peak shape ($Q$) behavior is reflective of energy dissipation in the tip–surface junction, such that a lower relative $Q$ value is indicative of a more viscous/compliant material. In summary, the results in Fig. 2 strongly suggest that the PS matrix is stiffer and less viscous than the PCL material, in agreement with previous works involving nanomechanical mapping by SPM techniques[48] of these samples.

This analysis based on the simple model of harmonic oscillator is extremely useful; however, as previously discussed, the reliance on a SHO type model precludes the investigation of subtle but important materials non-linearities (dynamic mechanical responses in compliant materials). Hence, further progress requires comprehensive data-driven approaches to visualize the data and elucidate underlying physics, ultimately yielding higher veracity functional imaging.

**Graph-Bootstrapping.** Over the past 2 decades, networks have become an invaluable tool for extracting insights from large-scale complex systems in many branches of science[49, 50]. Despite successful applications in numerous areas, network analytics usually require existing graph databases, which are constructed by manually labeling instances over a long time. However, of interest here is the emerging class of SPM techniques involving

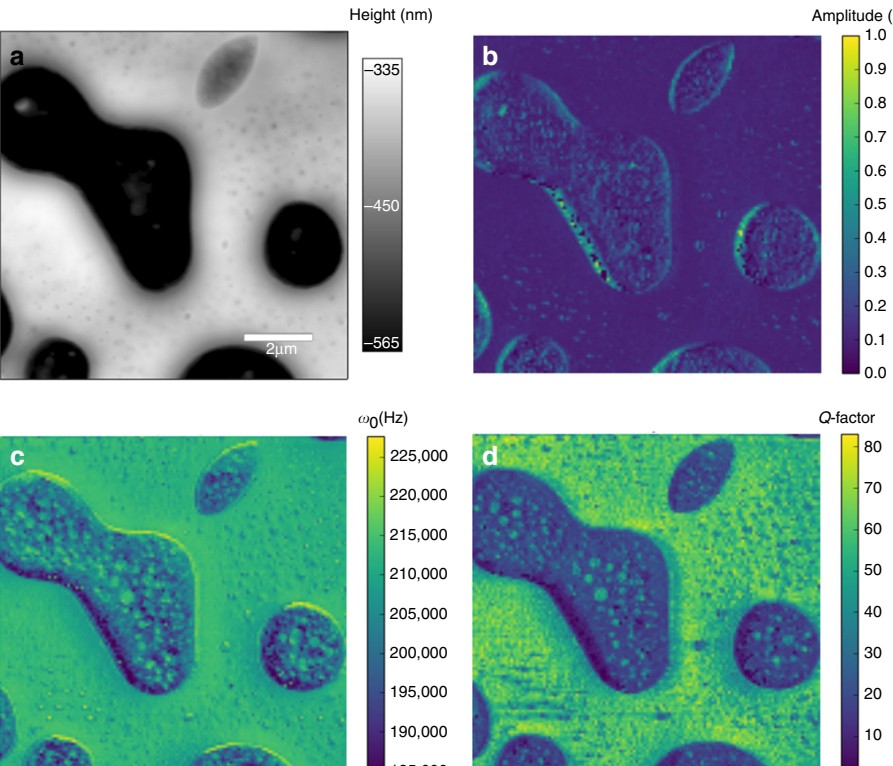

**Fig. 2** Traditional BE-SPM imaging on a thin-film polymer blend of PCL in a PS matrix. **a** Band excitation scanning probe microscopy (BE-SPM) topography of polystyrene/polycaprolactone (PS/PCL) sample. Spatial mappings of (**b**) amplitude, $A$, **c** contact resonance (CR) frequency, $\omega_0$, and **d** $Q$-factor, fitted via the simple model of harmonic oscillator (SHO)

significantly increasing levels of data capture, which often lack prior physical understanding of the tip–sample interactions and their relationship to particular material functionalities. Therefore, to gain insights into the behavior of both global and local relationships within high-dimensional measurements, we first construct the nearest neighborhood graph[51]. The complexity of constructing an exact nearest neighbor graph is $O(n^2 p)$, which is too expensive for high-dimensional datasets $\{X_1, X_2, \ldots, X_n\}$ where $X_i \in R^p$, especially for SPM datasets in which depending on the detection mode (Broad-band, G-Mode) $p$ can reach to $10^4$. Recently, Tang et al.[52] proposed LargeVis, a very efficient algorithm to build approximate nearest neighbor graph based on random projection tree[53] and neighbor-exploring[54] techniques. The weights of edges are calculated in a similar way to the distributed stochastic neighbor embedding (t-SNE)[55] method, by converting the Euclidean distances between neighbors into conditional probabilities that represent similarities. Furthermore, LargeVis layouts the nearest neighbor graph in low-dimensional manifold space following a principled probability model solved via asynchronous stochastic gradient descent[56].

For SPM datasets, especially for high-dimensional detection mode ($p \sim 10^4$), we found LargeVis projection on the low-dimension manifold space tends to be few bulk modules, which can usually distill most salient parts of the material. However, it is difficult to further explore local details in those modules. At the same time, the low-dimensional (2D/3D) coordinates calculated by LargeVis preserve all relationships between measurements. To explore intrinsic structures and present them in a clearer way, we propose reconstructing the graph based on the LargeVis low-

dimensional manifold coordinates and subsequently recalculating the manifold layout positions based on the reconstructed graph, following the same principled probability model. We refer to this method as Graph-Bootstrapping.

Traditionally, manifold embeddings are used for visualization purposes only, that is, to overlay known labels on the manifold points. Here, we expand this approach by learning manifold from high-dimensional measurements and subsequently clustering on the manifold to provide a natural way of visualizing and denoising the data, as well elucidating the relevant physics. This task however brings two main challenges. First, the manifold layout should consist of distinguishable groups which when accessible will enable straightforward clustering to be performed. Second, the clustering algorithm should be robust enough to cover irregular manifold layouts. Figure 3 presents a concrete example of manifold embedding and clustering with simulated Gaussian curves. Specifically, we consider four groups of parameters (Fig. 3a): two ranges of mean in [0, 1], [10,11] with the step size of 0.01, and two standard deviations: 0.2, 1. Each Gaussian curve has a unique combination of mean and standard deviation (Fig. 3b). We compare two popular visualization methods, Isomap[51] and t-SNE[55], with our proposed Graph-Bootstrapping (GB) method. For clustering, we use $K$-means and hierarchical density estimates method (HDBSCAN[57]), which is explained in detail in subsequent sections.

From Fig. 3, we can see that the Isomap method could not differentiate Gaussian curves of different standard deviations. For clustering, $K$-means works correctly on t-SNE and Graph-Bootstrapping manifolds, given the right tuning parameter, $K$,

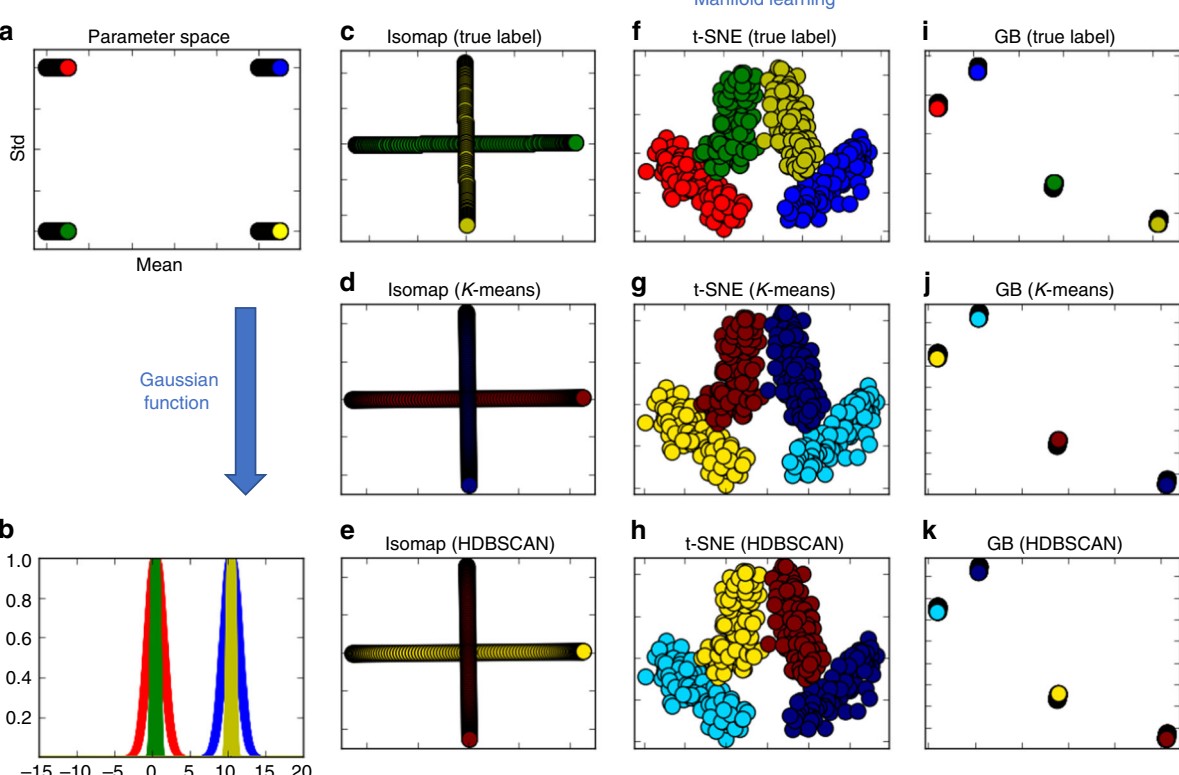

**Fig. 3** Illustration of manifold embedding and clustering on simulated Gaussian curves. **a** Four groups of parameters including two ranges of mean in [0, 1], [10,11] with the step size of 0.01, and two standard deviations of 0.2, 1. **b** Four groups of Gaussian curves, generated from the four groups of parameters in **a**. Each Gaussian curve has a unique combination of mean and standard deviation. **c–e** Manifold points estimated by Isomap[51], colored by true labels and clustered labels via $K$-means and hierarchical density estimates method (HDBSCAN[57]), respectively. **f–h** Manifold points estimated by t-SNE[55], colored by true labels and clustered labels via $K$-means and HDBSCAN, respectively. **i–k** Manifold points estimated by Graph-Bootstrapping (GB), colored by true labels and clustered labels via $K$-means and HDBSCAN, respectively

the number of clusters. However, the number of clusters is usually an unknown parameter, which in the case of SPM imaging is related to material properties that we would like to estimate. Thus, the manifold learning and clustering should share conjugate hyperparameters that are based on the local structure (such as nearest neighbors), which is the underlying logic behind the Graph-Bootstrapping presented herein.

**High-veracity BE-SPM Imaging.** Graph-Bootstrapping is firstly applied for analysis of the BE and broadband BE-SPM measurements of the polymer mixture. Graph-Bootstrapping reveals domain differentiation despite the fact that the algorithm is purely statistical in nature and does not require any prior information regarding the material and any differentiating structures. It is also an efficient approach. SHO fittings for a high-dimensional broadband excitation ($p = 15{,}159$) is prohibitive, yet Graph-Bootstrapping took 20 min to process 4GB of broadband BE-SPM measurements on a single workstation (Intel Xenon E5-1650V3, 32GB DDR3 RAM). Figure 4 compares the results by LargeVis and Graph-Bootstrapping. The relationship between LargeVis and Graph-Bootstrapping clusters is illustrated in following sections. Here we distinguish BE datasets, which contain a single CR peak (narrow band) with broadband resonance BE datasets that capture a broad frequency range spanning several resonance peaks.

Previous studies have shown that complex networks often exhibit hierarchical organizations[58–60]. Correspondingly, the geometry derived from bootstrapping clearly displays the hierarchical groups within SPM measurements, which allows us to utilize HDBSCAN[57, 61] to gain insights into many network properties, yielding the spatial mapping of heterogeneous local structures of material. Mathematically, HDBSCAN relies on the mutual reachability distance, which works in a conjugate way with the neighbor exploring stage during graph construction, that is, a

neighbor of a neighbor is also likely to be a neighbor: $D_{\mathrm{mreach},k}(a,b) = \max\{\mathrm{core}_k(a),\ \mathrm{core}_k(b),\ d(a,b)\}$, where $d(a,b)$ is the original metric distance between points $a$ and $b$, $\mathrm{core}_k(x)$ is the core distance of a point $x$ to cover its $k$ nearest neighbors. Supplementary Fig. 1 shows the relationship between the total cluster number and the minimum nearest neighbor number, $k$. HDBSCAN mostly yields two clusters with a few outliers of three clusters based on LargeVis manifolds (Supplementary Fig. 1). Since there is no clear trend in the resulting cluster number, we choose the majority of total cluster numbers, a value of 2. Supplementary Fig. 2 illustrates the spatial mappings of the two LargeVis clusters.

Compared to LargeVis, the total number of clusters based on the Graph-Bootstrapping manifold decreases continuously as $k$ increases (Supplementary Fig. 1). At the first glance, the resulting number of clusters $P(k)$, varies as a power of $k$:

$$P(k) = Ck^{-\alpha} \qquad (3)$$

where $\alpha$ is called the exponent of the power law that exists in vast branches of natural sciences, computer sciences, economics, and social sciences[62–67]. To get the fitting of power law distribution, we used the software developed by Alstott et al.[68]. We first normalized the cluster numbers and calculated the log–log plot of probability density function of the empirical data as well as fitted power law (Supplementary Fig. 1). We also fitted the trend with exponential distributions (Supplementary Fig. 1), another popular candidate for heavy-tailed distributions:

$$P(k) = Ce^{-(k-\mathrm{offset})/\tau} \qquad (4)$$

where $\tau$ is the relaxation constant. For exponential fitting, from $k \approx 3\tau$ (more accurately, it should be $k \approx 3\tau + \mathrm{offset}$, for simplicity, we omit the notation of offset in the rest of paper), the cluster number tends to be stable as $k$ increases.

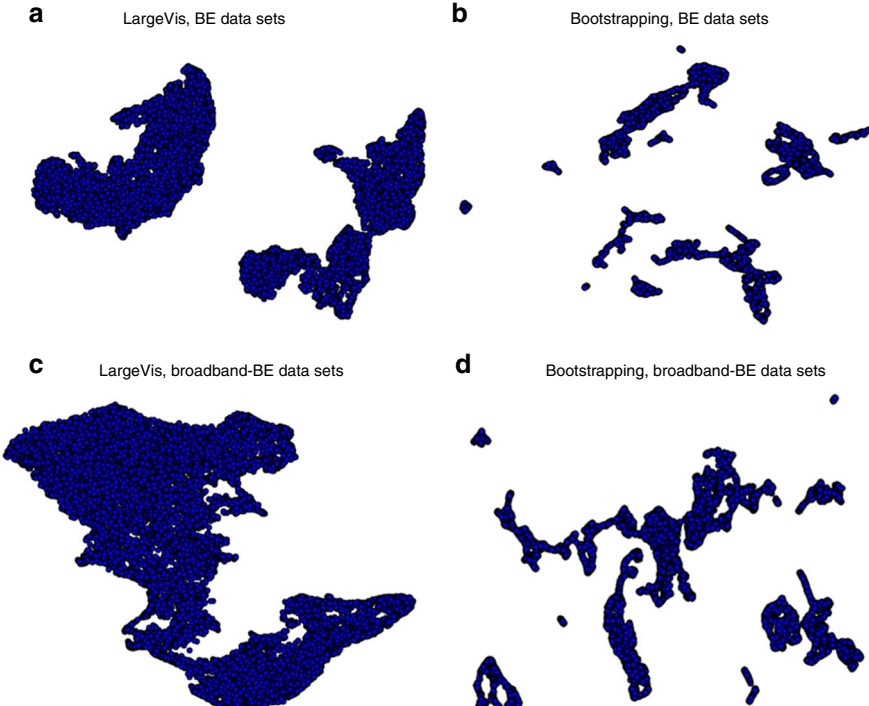

**Fig. 4** Manifold layouts by LargeVis and Graph-Bootstrapping calculated from the polymer mixture datasets. **a** Manifold points estimated by LargeVis, based on the band excitation scanning probe microscopy (BE-SPM) datasets. **b** Manifold points estimated via Graph-Bootstrapping by firstly reconstructing the graph from LargeVis manifold in **a** and then recalculating the manifold points based on the reconstructed graph. **c**, **d** The same analysis procedure with those in **a**, **b**, but applied on the broadband BE-SPM datasets collected on the same polymer mixture sample

To establish the quality of clustering, corresponding to the number of distinct surface functionalities, we calculated the (cumsum) standard deviations (STD) of SPM measurements within each cluster at different $k$ values. Supplementary Fig. 3 displays the boxplot of cluster STD distributions. Like Supplementary Fig. 1, we can see big jumps around $k \approx 3\tau$, $2\tau$, $\tau$. Clusters with lower STDs are more stable numerically. On one hand, the newly raised clusters could reveal subtler physical factors that affect materials at higher spatial resolution; on the other hand, the newly raised clusters could be essentially of the same material property, but are divided by SPM measurements noises. Supplementary Fig. 4 displays manifold clusters and the spatial distributions at low- and high-hierarchical levels. We note small inclusions (Supplementary Fig. 4) in the semicrystalline PCL and PS structure, which are likely smaller crystallites formed during the film synthesis, and are useful in highlighting the spatial resolution of the approach. What's more, we see newly bootstrapped clusters as $k$ decreases seem to be randomly located when $k$ is below $2\tau$

(Supplementary Fig. 4). The above observations are consistent with our conjecture on Supplementary Figs 1, 3. Combining Supplementary Figs. 1, 3, 4, we empirically propose $k \approx 3\tau$ to be the optimal choice between accuracy and overfitting. Or, if we could estimate the measurement noise level, then it may be a good reference. That is to say, we are less faithful in clusters whose standard deviations are less than that of system noise (if obtainable).

Based on above analysis, we can directly denoise the data, visualize it, and extract underlying physics parameters based on the manifold clusters in Supplementary Fig. 5, where the BE signals of bootstrapped clusters and their spatial mappings clearly distill the subtle material surface details, especially for the interface region. One can further calculate the similarity loadings by calculating pairwise distances between the mean SPM curve of the cluster and every raw SPM curve. The resulting similarity loadings (inversion of Euclidean distances) clearly highlight different material regimes (Supplementary Figs. 6, 7), yielding subtler and more enriched local heterogeneity than the SHO

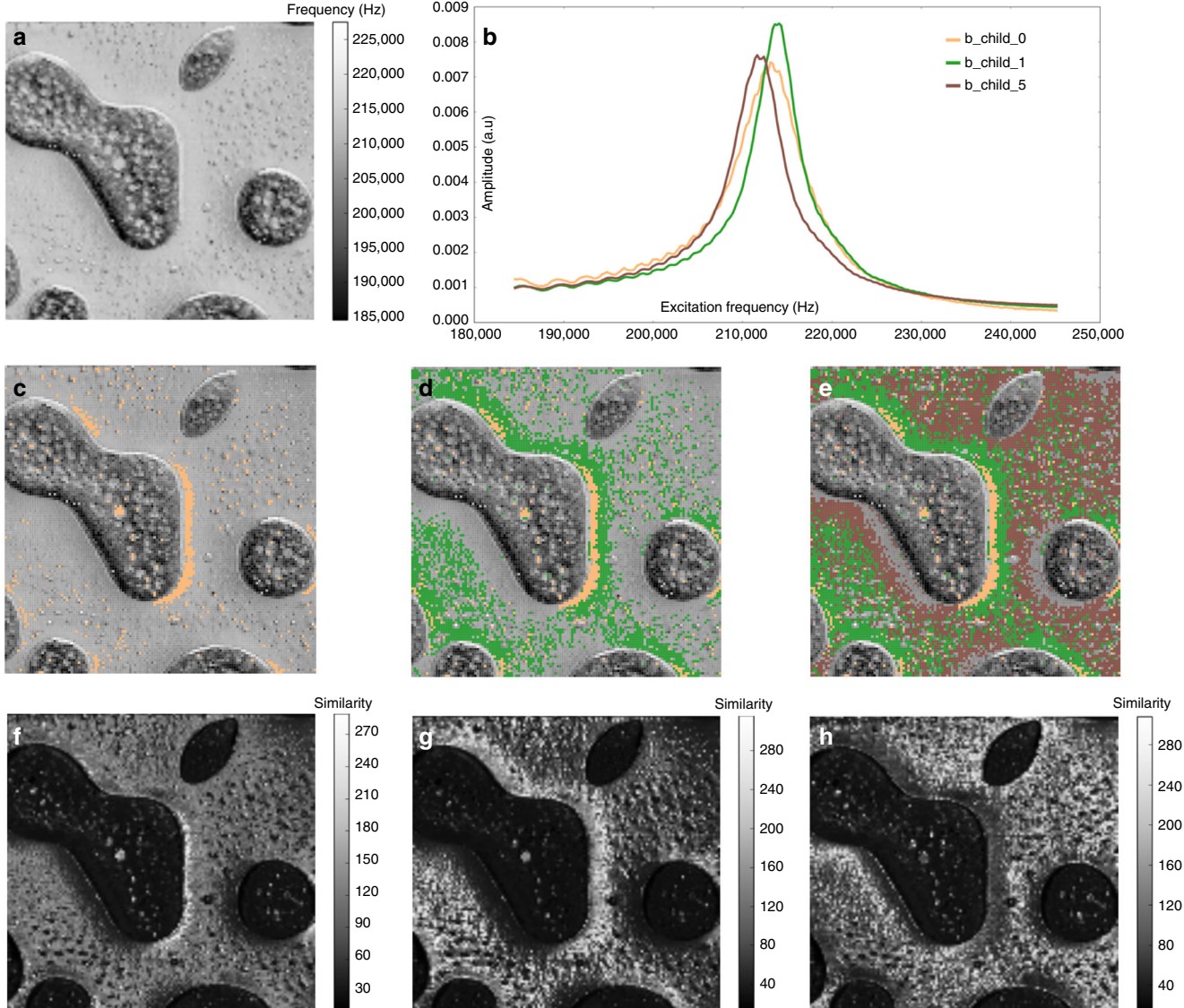

**Fig. 5** Application on the polymer mixture sample with band excitation scanning probe microscopy. **a** Spatial mapping of frequency via the simple model of harmonic oscillator (SHO) fitting. **b** Mean SPM response curves of three bootstrapped clusters, b_child_0, b_child_1, and b_child_5. **c–e** Cluster labels (b_child_0, b_child_1, b_child_5) cumulatively overlaid on SHO fitted frequency map. **f–h** Similarity loadings of clusters b_child_0, b_child_1, and b_child_5, respectively

fitting. Those similarity loadings can verify the accuracy of the Graph-Bootstrapping method.

In Fig. 5, we present deeper insights obtained via Graph-Bootstrapping. Figure 5a is the spatial mapping of frequency from SHO fittings. Such maps are normally transformed using analytical models into material property maps such as elastic modulus. The brighter area (higher frequency) in Fig. 5a corresponds to the PS matrix of higher stiffness. Noticeably, little structural inhomogeneities are detected within the PS matrix based on Fig. 5a (or in other SHO fitting parameters as shown in Fig. 2). Meanwhile Graph-Bootstrapping illustrates at least three distinguishable clusters (Fig. 5c–e) within the PS matrix with the corresponding SPM response curves (Fig. 5b). To quantitatively check the accuracy of clustering, we provide the similarity loadings in Fig. 5f–h. We do see the similarity loading has higher similarity values at the spatial positions of the cluster. In Fig. 5b, we see the nonlinear ring patterns at lower frequency range which are ignored by definition in the SHO fittings. In this way, our generalized method can be used to unveil hidden details, without the need for an a priori model. We further note the mean SPM curve of cluster b_child_1 has a higher height than the other two clusters, which SHO fittings failed to reveal. We also attached Graph-Bootstrapping results of broad-band BE datasets collected on the same polymer mixture sample in Supplementary Figs. 8, 9, 10, 11.

**Application on 3D-AFM Imaging.** To further demonstrate the broad capability of Graph-Bootstrapping, we applied it to the 3D-AFM dataset that has been recently investigated by Söngen et al.[69] to resolve point defects in the hydration structure of calcite (10.4). For each $(x, y)$ surface position, excitation frequencies extracted at different $z$ piezodisplacements (corresponding to various hydration layers) are inputted to the Graph-Bootstrapping.

Figure 6a is the manifold clustering results of the first-order bootstrapping, and correspondingly Fig. 6b is the surface distribution of cluster labels revealing the lattice structure. We performed hierarchical clustering on the second-order bootstrapped manifolds (Fig. 6d) and overlaid the same set of labels onto the first-order manifolds (Fig. 6c) where we can see clusters still aggregate. Figure 6e shows the atomic structure of the (10.4) surface unit cell with a scale bar that applies to all surface spatial mappings. Figure 6f displays the surface positions of the second-order bootstrapped clusters revealing subtle details between the Ca site and the carbonate site, which can be further investigated via the excitation frequency shift profiles of the clusters presented in Fig. 6g. First, we note that the shift curve of cluster_5 consistently exhibits the largest local maxima and the smallest local minima in the third, fourth, and fifth hydration layer, corresponding to the profile of the Ca defect site which has been verified manually by Söngen et al.[69]. Second, Graph-Bootstrapping also elucidated the irregular flattening of the cluster_0 curve between the fourth and fifth hydration layer as well as its large shifts of local maxima and local minima between the second and fourth hydration layer. We can assure this irregularity by comparing surface label sites and similarity loading (Supplementary Fig. 12). We can see that the similarity loading indeed has higher similarity values at the spatial positions of the cluster_0.

## Discussion

Over the past 10 years, several groups have made significant contributions towards the quantification of viscoelastic properties by CR SPM[70–72]. However, even CR-SPM still incessantly undergoes developments to this day, in both modeling and hardware. This is because there is still a lack in knowledge of how the cantilever precisely behaves on certain materials in the

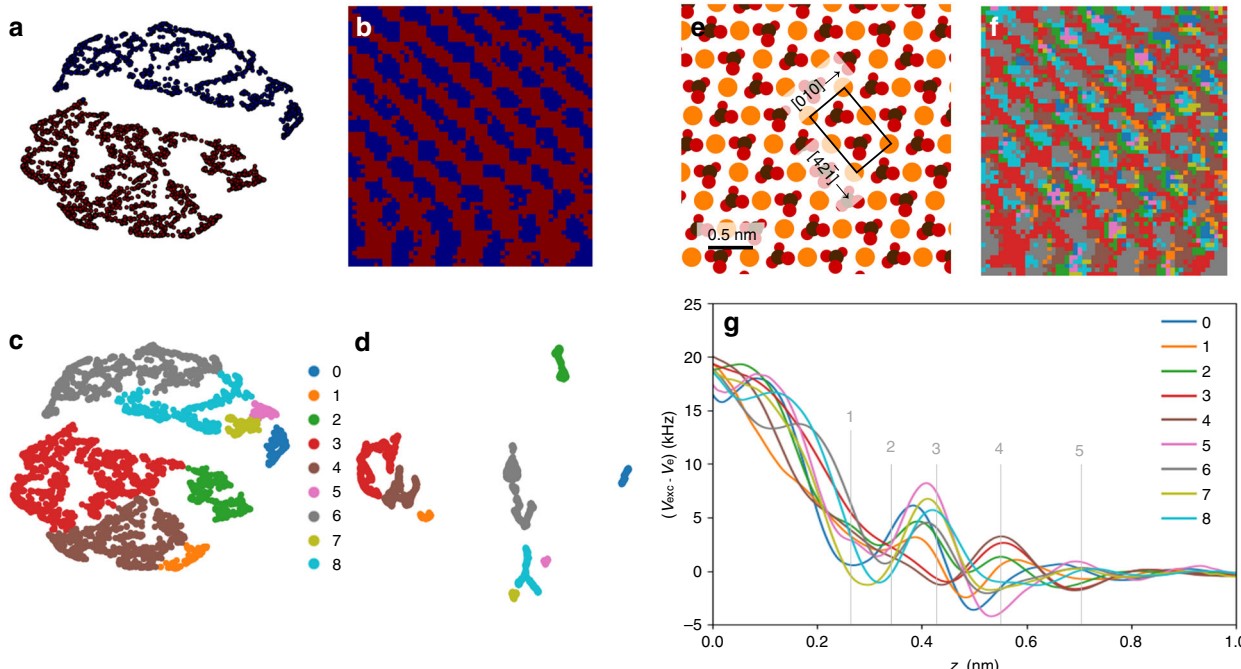

**Fig. 6** Application on hydration structure of calcite with 3D-AFM. **a** First-order bootstrapping manifold colored by clustering labels, that is, the manifold derived from the reconstructed graph based on LargeVis manifold and **b** the spatial mapping of the corresponding labels. **c, d** Comparison of manifold patterns (first- and second-order bootstrapping) colored by the same set of clustering labels where hierarchical clustering was performed on the second-order bootstrapping manifold. Second-order bootstrapping manifold is derived from the reconstructed graph based on the first-order manifold. **e** Atomic structure of the calcite (10.4) surface unit cell, reproduced from Ref.[69]. (Copyright [2018], APS). **f** Surface positions and **g** excitation frequency shift profiles of second-order bootstrapping clusters

presence of background forces. We proposed a generalized method to spatially map the local inhomogeneities on the high-dimensional datasets captured in these modes, and to do so without requiring an a priori model or placing any constraints on the data. As we highlight in Fig. 5, even at this stage, we can pinpoint deviations from expected cantilever behavior that are not described by a simple model adopted as the gold standard. This information, in turn, can be used to evaluate the effectiveness of existing approaches. After extracting patterns revealed by Graph-Bootstrapping, further efforts can be made on deeper theoretical and experimental study on correlations between the whole shape of SPM curves and material properties.

To further illustrate the broad capability of Graph-Bootstrapping, we applied it to 3D-AFM and demonstrated its successes on resolving irregular hydration structure of calcite at atomic resolution. In summary, we develop a universal algorithm for representation of high-dimensional SPM data. Based on the combination of network analysis and search for low-dimensional manifold, this approach reveals and hierarchically represents salient material features with rich local details, further enabling denoising, classification, and high-resolution functional imaging. While demonstrated for BE-SPM and 3D-AFM, this approach can be universally applicable to other data-rich imaging methods. We further pose that learning of low-dimensional manifold representing the data will open the pathway for extraction of the relevant physics coupled between material and imaging systems.

## Methods

**Band-excitation scanning probe microscopy.** Mechanical measurements were performed on thin-film polymer blend samples consisting of PCL inclusions embedded in a PS matrix. Samples were prepared by spin coating in 2:1 ratio as described in previous work by Kocun et al.[48]. Film thickness was >300 nm. Mechanical properties of the sample were previously measured using bimodal tapping mode[48, 73] where Young's modulus of the PS was assumed to be 3.0 GPa and used as a reference to determine the stiffness of PCL (~0.85 GPa).

All measurements on polymer sample were performed using a commercial AFM system (Cypher, Asylum Research and Oxford Instruments Company), which was equipped with a laser for photothermal excitation (Bluedrive module) and used for mechanical perturbation of the AFM tip–sample interaction. CR-AFM measurements were performed in contact mode where the tip was held in constant force (~65 nN) with the surface using static deflection feedback. BE-AFM imaging was operated at normal AFM imaging speeds (1 Hz), and the AFM system was coupled with data acquisition and arbitrary waveform generators (National Instruments, NI5122 and NI5412), which were controlled using custom Matlab code (MathWorks). Measurements used a gold-coated NSC36 (Micromasch) cantilever with a calibrated spring constant of 0.89 N/m and free air resonance frequency of 44.7 kHz. The BE excitation center frequency, which was used to excite the photothermal laser position on the AFM cantilever, was chosen to approximately match the CR frequency ($\omega_0 = \sim 210$ kHz) and a bandwidth ~100 kHz using a sampling rate of 4 MHz leading to a data capture of ~20 bins per band. In addition, broadband excitation/detection was implemented, using a center frequency of ~500 kHz and bandwidth of ~490 kHz allowing a larger number of resonance peaks to be captured simultaneously.

**Three-dimensional atomic force microscopy (3D-AFM).** 3D-AFM measurements on calcite were performed by Söngen et al.[69] and we refer the reader to this text for additional details.

**Code availability.** The Graph-Bootstrapping codes used in this study can be made available from the corresponding authors upon request.

**Data availability.** The data that support the findings of this study are available from the corresponding authors upon request.

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

## Acknowledgments

The research was performed at the Center for Nanophase Materials Sciences, a U.S. Department of Energy, Office of Science User Facility at Oak Ridge National Laboratory. T.F. and K.M. acknowledge support from the World Premier International Research Center Initiative (WPI), MEXT, Japan. The authors gratefully acknowledge Hagen Söngen and Angelika Kühnle for help and discussion on 3D-AFM datasets.

## Author contributions

X.L. and S.V.K. conceived and designed the study. X.L. developed Graph-Bootstrapping methodology and performed the associated data analysis. L.C. and S.J. performed SHO fitting analysis on BE-SPM datasets. K.M. and T.F. provided 3D-AFM measurements of calcite. S.V.K. and S.J. supervised the study. X.L. and S.V.K. co-wrote the article with comments from all authors.

## Additional information

**Competing interests:** The authors declare no competing interests.

