## [Peer Review File · Nature Communications]

Reviewers' comments:

Reviewer #1 (Remarks to the Author):

The paper proposes a technique for computing low dimensional representations of full spectral response data collected through scanning probe microarrays (SPM). SPM has become the state-of-the-art of imaging and visualizing nanoscale surface structures. SPM usually generates high-veracity, high-dimensional response signals, where the true patterns reside in an intrinsic low-dimensional space. This provides a unique use case scenario for the recently developed computational methods that learn low-dimensional embeddings of large-scale, high-dimensional data. The main contribution of this paper is the proposal of a graph-bootstrapping method that is built on top of a graph embedding method and applied to computing the low dimensional representation of SPM signals.

Significance: SPM has become a standard tool for imaging and exploring nano-scale structures, which is applied to many fields and many problems such as material sciences, nano-biology, genetics, electrochemistry, and semiconductors. Any significant development in analyzing SPM signals may result in an advancement of these fields. Comparing to conventional mono-frequency SPMs in which the cantilever is made to vibrate at a specific frequency, multifrequency SPMs has significant advantages and has become the state-of-the-art. The downside, however, is that multifrequency SPMs generates large-scale, high-dimensional signals that makes visualization and exploration hard. The method proposed by this paper addresses this fundamental problem. If it solves this problem well, the proposed method is likely to generate a considerable impact. With that said, the authors are expected to provide more quantitative evidence on the significance of the proposed method compared to the current state-of-the-art. I understand a quantitative evaluation of a visualization/exploration task may be hard. One suggestion is to generate many simulated cases like Figure 3 and calculate the accuracy of each method.

Methodology: multifrequency SPMs generate large-scale, high-dimensional signals, which are intrinsically generated from non-linear transformations of a low-dimensional (e.g., 2D or 3D) surface. This creates a perfect use case of non-linear, low-dimensional embedding learning methods such as t-SNE and LargeVis. The proposed graph bootstrapping method is technically sound, which is built upon LargeVis but explores a bootstrapping procedure that iteratively updates the low-dimensional embedding and the manifold structure. The manifold structure constructed after graph bootstrapping yields much better cluster structure through a standard HDBSCAN procedure. While the procedure is sound, more detailed descriptions should be provided on how graph bootstrapping is done, and why it works. For example, how many iterations of bootstrapping is needed, and whether every iteration embeds the manifold structure into the same number of dimensions? Intuitively, the reason graph bootstrapping works is because of its interwoven procedure of graph embedding and denoising. If this is the case, experiments should be provided to demonstrate (and quantify) the dynamics of patterns and noises over the iterations.

Evaluation: The paper provides quantitative evaluation on synthetic data and mostly qualitative evaluation on real data. While quantitatively evaluating data visualization and exploration is hard, I suggest the authors report quantitative evaluation on more complicated synthetic data (e.g., arbitrary shapes) or real data where the ground-truth structural patterns (e.g., clusters) are annotated.

Conclusion: The paper proposes a generic computational method that effectively visualizes and analyzes the multi-dimensional signals of multi-frequency SPMs. The method, based on a bootstrapping procedure of manifold construction and graph embedding, is able to more effectively reveal the cluster structures of the surface on a low dimensional space. This method, if the author can better justify the significance of its advantage over the current art, can be potentially applied to many fields and

problems that rely on the use of SPMs.

Reviewer #2 (Remarks to the Author):

The Ms. reports an approach based on the low-dimensional simplification of the high-dimensional data generated by band excitation probe microscopy via a combination of graph analytics and hierarchical clustering. The authors perform some measurements on a polymer blend to illustrate their points. The authors' have pioneered several sophisticated scanning probe methods such as band excitation and G-mode. They also have a deep understanding on data analysis such as principal component analysis, machine learning and other state-of-the-art big data tools.

The introduction and several sections of the Ms. are written in an accessible style for the non-specialist in data analysis.

In terms of quantitative imaging, the spatial resolution and quantitative mapping are below the state-of-the-art in SPM imaging of polymers. It seems that the authors have aimed to fix some of the technical limitations of BE AFM without demonstrating any relevant application in terms of material properties. This Ms. is rather technical for NCOMMS. It should be submitted to a more specialized journal.

Reviewers' comments:

Reviewer #1 (Remarks to the Author):

The paper proposes a technique for computing low dimensional representations of full spectral response data collected through scanning probe microarrays (SPM). SPM has become the state-of-the-art of imaging and visualizing nanoscale surface structures. SPM usually generates high-veracity, high-dimensional response signals, where the true patterns reside in an intrinsic low-dimensional space. This provides a unique use case scenario for the recently developed computational methods that learn low-dimensional embeddings of large-scale, high-dimensional data. The main contribution of this paper is the proposal of a graph-bootstrapping method that is built on top of a graph embedding method and applied to computing the low dimensional representation of SPM signals.

We thank the reviewer for thorough examination of the manuscript.

Significance: SPM has become a standard tool for imaging and exploring nano-scale structures, which is applied to many fields and many problems such as material sciences, nano-biology, genetics, electrochemistry, and semiconductors. Any significant development in analyzing SPM signals may result in an advancement of these fields. Comparing to conventional mono-frequency SPMs in which the cantilever is made to vibrate at a specific frequency, multifrequency SPMs has significant advantages and has become the state-of-the-art. The downside, however, is that multifrequency SPMs generates large-scale, high-dimensional signals that makes visualization and exploration hard. The method proposed by this paper addresses this fundamental problem. If it solves this problem well, the proposed method is likely to generate a considerable impact.

We again thank the reviewer for the appreciation of the significance of our proposed method. In our ongoing work about electrochemistry, Graph-Bootstrapping has elucidated material-dependent, timing-resolved patterns, inspiring new thoughts on physical interpretations of SPM signals. Graph-Bootstrapping has also been successfully applied to another data-rich imaging method, 4D electron diffraction, which was used in this reply as another numerical example to further illustrate the working mechanism behind Graph-Bootstrapping. In the following, we have attempted to respond to all concerns. In the end, we summarize our analysis based on Figure R9 to highlight the significance of its advantage over the current state-of-the-art.

With that said, the authors are expected to provide more quantitative evidence on the significance of the proposed method compared to the current state-of-the-art. I understand a quantitative evaluation of a visualization/exploration task may be hard. One suggestion is to generate many simulated cases like Figure 3 and calculate the accuracy of each method.

The reviewer makes a critical point about quantitative evaluation of clustering. To address this issue, in supplementary file (supplementary Figures 2,3) we provided the **similarity loadings** for Mean SPM measurements of clusters. For each cluster, we first calculate the Mean contact resonance curve. Then we calculate the inversion of pair-wise Euclidean distance between the cluster Mean curve and the curve at each pixel location. To check the clustering accuracy quantitatively, we can compare the spatial distribution of cluster label and the corresponding similarity loading map. Figure R1 are the spatial distributions of cluster labels (denoted in color red) and similarity loading maps.

Figure R1: Spatial distributions of cluster labels (denoted in color red) and similarity loading maps.

On manifold embedding algorithms, LargeVis exceeds T-sne in terms of stability and efficiency, considered as the new state-of-the-art. For example, with BE datasets of size (128*128,249), the standard T-sne function provided in Python sklearn package caused the kernel crash. Meanwhile, built upon LargeVis, Graph-Bootstrapping took less than 20 minutes to process a much higher dimensional broad-band BE datasets of size (128*128, 15159) on the same computer, yielding a much better cluster structure than LargeVis alone (Figure 4 in manuscript and Supplementary Figures 4,5,6,7).

The mainstream data mining method in SPM literature is PCA. Figure R2 are the PCA loading maps.

Figure R2: PCA loadings.

Figure R1 and Figure R2 speak for themselves. What's more, PCA only gives the abstract components of SPM curves meanwhile Graph-Bootstrapping simultaneously yields characteristic (Mean) SPM curves with similarity loadings. Any future solutions can be easily applicable as an additional step.

Methodology: multifrequency SPMs generate large-scale, high-dimensional signals, which are intrinsically generated from non-linear transformations of a low-dimensional (e.g., 2D or 3D) surface. This creates a perfect use case of non-linear, low-dimensional embedding learning methods such as t-SNE and LargeVis. The proposed graph bootstrapping method is technically sound, which is built upon LargeVis but explores a bootstrapping procedure that iteratively updates the low-dimensional embedding and the manifold structure. The manifold structure constructed after graph bootstrapping yields much better cluster structure through a standard HDBSCAN procedure. While the procedure is sound, more detailed descriptions should be provided on how graph bootstrapping is done, and why it works. For example, how many iterations of bootstrapping is needed, and whether every iteration embeds the manifold structure into the same number of dimensions? Intuitively, the reason graph bootstrapping works is because of its interwoven procedure of graph embedding and denoising. If this is the case, experiments should be provided to demonstrate (and quantify) the dynamics of patterns and noises over the iterations.

The reviewer makes an exciting point here. As the reviewer realized, SPM provides a perfect application scenario that could stimulate the development of manifold embedding methods for large-scale, high-dimensional data. We kept all the tuning parameters constant throughout the iterations, such as dimension of the manifold and the number of nearest neighbor, K . We believe that a key principle behind Graph-Bootstrapping's success is that the graph construction and HDBSCAN work in a conjugate way based on local-linkage-structure: a neighbor of a neighbor is also likely to be a neighbor. A key stage in graph construction is the Neighbor Exploration step:

1. Create the Max heap H_i for each node i in the graph.
2. For each neighbor node j of node i , calculate pairwise distance between node i and each neighbor node l of node j , $\text{dist}(i,l)$
3. Push l with $\text{dist}(i,l)$ into H_i
4. Pop if H_i has more than K nodes

Correspondingly, HDBSCAN relies on the *mutual reachability distance*:

$$D_{\text{mreach},k}(a,b) = \max\{\text{core}_k(a), \text{core}_k(b), d(a,b)\},$$

where $d(a,b)$ is the original metric distance between points a and b , $\text{core}_k(x)$ is the *core distance* of a point x to cover its K nearest neighbors. We can see that the mutual reachability distance conveys a similar philosophy with that in Neighbor Exploration stage.

An intriguing question is how many times we need bootstrapping. On SPM datasets, empirically we conjectured that bootstrapping once is enough to unveil enriched local-structure details without potential risk of over-clustering, together with our proposed tuning strategy based on Figure 5(d), Figure 7 and Figure 8 in the manuscript.

Figure R3 illustrates the dynamics of manifold patterns over iterations. HDBCAN was performed on bootstrapped manifolds. We overlaid the same set of cluster labels on both LargeVis and Bootstrapped manifolds. We can see that clusters derived in bootstrapped manifold are still aggregated on LargeVis manifold and none of them are scattered out.

Figure R3: Dynamics of manifold patterns over iterations.

For the more complex system in 4D STEM diffraction datasets of a graphene sample, i.e., each scanned pixel location is associated with a diffraction image-Ronchigram, high-order bootstrapping can indeed help to unveil heterogeneity structure existed in deeper hierarchy. For the clean synthetic datasets of high quality, LargeVis manifold amazingly fall into shape of a hexagon (Figure R4(a)). One can simply do spectral clustering with 7 clusters based on visual intuition that leads to unveiling polarization patterns in Ronchigram on graphene sublattices shown in Figure R5.

Figure R4: (a) Spectral clustering results on LargeVis manifold from synthetic Ronchigram datasets and (b,c,d) HDBSCAN clustering labels on LargeVis, first-order and second-order bootstrapping manifolds derived from experimental Ronchigram datasets. HDBSCAN clustering was performed on second-order bootstrapping manifold.

Figure R5: Synthetic data analysis. (a,c) Spatial distributions of cluster labels over graphene two sublattices. (b) Mean and STD of cluster Ronchigram

However, for larger experimental datasets, we more often could not get a good guess on the number of clusters based on LargeVis manifold shape due to noise and distortions associated in the experimental data (imagine Figure R4(b) without those cluster labels). We performed bootstrapping twice. We did HDBSCAN on second-order bootstrapping manifold and overlaid the same set of cluster labels on manifolds of all three iterations (Figure R4 (b,c,d)). Similar with Figure R3, clusters derived in second-order bootstrapping manifold are still aggregated on manifolds of previous iterations. Another twist in experimental datasets is that the deflection pattern in Ronchigram is too shallow to identify by human eye. Figure R8 (a) is the corresponding HADDF image and (b) is an individual Ronchigram image. We then refer to the power of **similarity loadings**. Based on Figure R5, we grouped the similarity loadings from synthetic datasets into sublattices A and B, shown in Figure R6. Then based on our findings in Figure R6, we also grouped the similarity loadings from experimental datasets into sublattices A and B, shown in Figure R7. Finally, based on Figure R7, in Figure R8 (c,d) we separately overlaid sublattices' labels on the HADDF image. We can see cluster labels are consistent with atom positions in the HADDF image.

Figure R6: Synthetic data analysis. (a,b,c) Similarity loadings of clusters over sublattice A. (d,e,f) Similarity loadings of clusters over sublattice B.

Figure R7: Experimental data analysis. (a-d) Similarity loadings of clusters over sublattice A. (e-h) Similarity loadings of clusters over sublattice B.

Figure R8: Experimental data analysis. (a) The HAADF image of graphene. (b) An experimental Ronchigram image. (c) Sublattice A labels overlaid on HAADF image. (d) Sublattice B overlaid on HAADF image.

Evaluation: The paper provides quantitative evaluation on synthetic data and mostly qualitative evaluation on real data. While quantitatively evaluating data visualization and exploration is hard, I suggest the authors report quantitative evaluation on more complicated synthetic data (e.g., arbitrary shapes) or real data where the ground-truth structural patterns (e.g., clusters) are annotated.

Conclusion: The paper proposes a generic computational method that effectively visualizes and analyzes the multi-dimensional signals of multi-frequency SPMs. The method, based on a bootstrapping procedure of manifold construction and graph embedding, is able to more effectively reveal the cluster structures of the surface on a low dimensional space. This method, if the author can better justify the significance of its advantage over the current art, can be potentially applied to many fields and problems that rely on the use of SPMs.

We thank the reviewer for constructive suggestions. We summarize above analysis in Figure R9.

Figure R9: Illustration of significance of Graph-Bootstrapping: (a) Spatial mapping of frequency from SHO fitting. (b) Mean response curves of bootstrapped clusters. (c-e) Cluster labels cumulatively overlaid on SHO fitted frequency map. (f-h) Similarity loadings of cluster b_child_0, b_child_1 and b_child_5.

Figure R9(a) is the spatial mapping of frequency from SHO fittings. The brighter area (higher frequency) in Figure R9(a) corresponds to polystyrene (PS) matrix of higher stiffness. Noticeable, little structural inhomogeneity's are detected within the PS matrix based on Figure R9(a) (or in other SHO fitting parameters as shown in Figure 2 of manuscript). Meanwhile Graph-Bootstrapping illustrates at least 3 distinguishable clusters within the PS matrix (Figure R9(c-e), Supplementary Figure 1) with corresponding response curves (Figure R9(b)). To quantitatively check the accuracy of clustering, we again provide the *similarity loadings* in Figure R9(f-h), by computing pair-wise similarity (inversion of distance) between Mean contact resonance curve for each cluster and the entirety of curves in the full dataset. We do see that similarity loading has much higher similarity values at the spatial positions of the cluster. Now we emphasize the contribution of Graph-Bootstrapping made on elucidating new understanding of imaging physics and correlation with material properties. In Figure R9(b), we see the nonlinear ring patterns at lower frequency range which are ignored by definition of the simple harmonic model used in fitting the data. In this way, our generalized method can be used to unveil hidden details, without the need for a priori model. We further note Mean contact resonance curve of cluster b_child_1 has a higher height than the other two clusters, which SHO fittings failed to reveal (Figure R9(a) and Figure 2 in manuscript). The analysis based on contact resonance height and correlations within the whole shape of contact resonance curve are key to unveil material-dependent, timing-resolved trends in our ongoing electrochemistry applications, which would not be possible without invention of Graph-Bootstrapping. To our best knowledge, there is a relative lack of physical interpretation on the height of contact resonance curve, compared with CR-Frequency and Quality Factor. After extracting patterns revealed by Graph-Bootstrapping, further efforts can be made on deeper theoretical and experimental study on correlation between the whole shape of contact resonance curve and quantitative material property. We replace the original Figure 9 in manuscript with Figure R9 and put the old one as Supplementary Figure 1.

We hope above replies can give the reviewer a better understanding of working mechanism of Graph-Bootstrapping, its advantages over current art as well as its potential for a broad range of functional imaging applications.

Reviewer #2 (Remarks to the Author):

The Ms. reports an approach based on the low-dimensional simplification of the high-dimensional data generated by band excitation probe microscopy via a combination of graph analytics and hierarchical clustering. The authors perform some measurements on a polymer blend to illustrate their points.

The authors' have pioneered several sophisticated scanning probe methods such as band excitation and G-mode. They also have a deep understanding on data analysis such as principal component analysis, machine learning and other state-of-the-art big data tools.

The introduction and several sections of the Ms. are written in an accessible style for the non-specialist in data analysis.

In terms of quantitative imaging, the spatial resolution and quantitative mapping are below the stat-of-the art in SPM imaging of polymers. It seems that the authors have aimed to fix some of the technical limitations of BE AFM without demonstrating any relevant application in terms of material properties. This Ms. is rather technical for NCOMMS. It should be submitted to a more specialized journal.

We thank the reviewer for the careful assessment of our work. We are glad our writing made the reviewer understand the method easily. In this manuscript, we develop a principally new approach for visualizing materials structure and functional properties from scanning probe microscopy (SPM) data, a key objective of force-based SPM techniques, based on similarity network analysis. As the reviewer #1 justified, “*SPM has become a standard tool for imaging and exploring nano-scale structures, which is applied to many fields and many problems such as material sciences, nano-biology, genetics, electrochemistry, and semiconductors. Any significant development in analyzing SPM signals may result in an advancement of these fields. Comparing to conventional mono-frequency SPMs in which the cantilever is made to vibrate at a specific frequency, multifrequency SPMs has significant advantages and has become the state-of-the-art. The downside, however, is that multifrequency SPMs generates large-scale, high-dimensional signals that makes visualization and exploration hard. The method proposed by this paper addresses this fundamental problem.*”

Over the past 10 years, several groups have made significant contributions towards quantification of viscoelastic properties by contact resonance SPM. However, even CR-SPM still incessantly undergoes developments to this day, in both modelling and hardware. This is because there is still a lack in knowledge of how the cantilever precisely behaves on certain materials, in the presence of background forces. It is not the purpose of this work to tackle these issues, although any future solutions can be easily applicable as an additional step, our purpose is instead to spatially map the local inhomogeneities on the high dimensional datasets captured in these modes, and to do so without requiring a priori model or placing any constraints on the data. As we highlight in the submitted manuscript even at this stage we can pinpoint deviations from expected cantilever behavior which are not described by simple model adopted by the gold standard. This information in turn can be used to evaluate the effectiveness of existing approaches.

While demonstrated here for band excitation SPM on a simple two-type polymer mixture sample, this approach is universal and can be generally applicable to other data-rich imaging methods. We further pose that learning of low-dimensional manifold representing the data will open the pathway for extraction of the relevant physics of imaging systems and will stimulate the next round of progress in SPM communities, boosting scientific discovery in a broad range of disciplines, such as energy storage, nanomaterial and biosystems. Please refer to our replies to Reviewer #1 for a better understanding of working mechanism of Graph-Bootstrapping and its advantages over state-of-the art.

Reviewers' comments:

Reviewer #1 (Remarks to the Author):

Thanks for providing the very detailed response! It did help me better understand the significance of the proposed method and its advantage over existing methods, such as T-SNE. Many of the figures and discussions in the response letter are very helpful for the readers to digest the method and results of the study, and I suggest the authors include some of them into the main manuscript and the rest as supplementary material. For example, the detailed algorithm of graph construction and HDBSCAN (on page 4 of response letter) will be very useful for readers to reproduce the results.

Reviewer #2 (Remarks to the Author):

I have read with interest the revised Ms. and the rebuttal letter. The technical quality of the Ms. is very high. It should be published in a technical journal. There is a mismatch between the points/claims of the authors and the experimental findings. The authors should have applied their methodology to address/solve a 'relevant problem' in nanoscale characterization.

Reviewer #1 (Remarks to the Author):

Thanks for providing the very detailed response! It did help me better understand the significance of the proposed method and its advantage over existing methods, such as T-SNE. Many of the figures and discussions in the response letter are very helpful for the readers to digest the method and results of the study, and I suggest the authors include some of them into the main manuscript and the rest as supplementary material. For example, the detailed algorithm of graph construction and HDBSCAN (on page 4 of response letter) will be very useful for readers to reproduce the results.

We are glad that reviewer got a better understanding of Graph-Bootstrapping. We thank reviewer for the suggestions and we have added discussions on graph construction and HDBSCAN under the section “Hierarchical Clustering on Manifold Space”.

Reviewer #2 (Remarks to the Author):

I have read with interest the revised Ms. and the rebuttal letter. The technical quality of the Ms. is very high. It should be published in a technical journal. There is a mismatch between the points/claims of the authors and the experimental findings. The authors should have applied their methodology to address/solve a ‘relevant problem’ in nanoscale characterization.

We really appreciate the suggestion of solving a ‘relevant problem’ in nanoscale characterization. To further demonstrate the broad capability of Graph-Bootstrapping and the associated uptake for the whole SPM community, we applied it to the 3D-AFM dataset that has been recently investigated by Söngen et al¹ to resolve point defects in the hydration structure of calcite (10.4) at the atomic resolution. The detailed analysis can be found under section “Application on atomic-resolution 3D-AFM” of the updated manuscript

Remarkably, we note that the shift curve of cluster_5 (Figure 10g) consistently exhibits the largest local maxima and the smallest local minima in the third, fourth and fifth hydration layer, corresponding to the profile of the Ca defect site which has been verified manually by Söngen et al¹. What’s more, Graph-Bootstrapping also elucidated the irregular flattening of cluster_0 curve between the fourth and fifth hydration layer as well as its large shifts of local maxima and local minima between the second and fourth hydration layer. We can assure this irregularity by comparing surface label sites and the similarity loading (Supplementary Figure 8a,b). We can see that the similarity loading indeed has higher similarity values at the spatial positions of the cluster_0.

While demonstrated for high-velocity mechanical mapping on a mixed polymer thin film via BE-SPM and resolving irregular hydration structure of calcite at atomic resolution by 3D-AFM, Graph-Bootstrapping can be universally applicable to other data-rich imaging methods, opening the pathway for extraction of the relevant physics coupled between material and imaging systems.

References:

1. Söngen, H. et al. Resolving Point Defects in the Hydration Structure of Calcite (10.4) with Three-Dimensional Atomic Force Microscopy. Phys. Rev. Lett. 120, 116101 (2018).

REVIEWERS' COMMENTS:

Reviewer #2 (Remarks to the Author):

This version includes data from Söngen et al. (PRL 120,116101 (2018)) that shows an atomically - resolved image of a calcite surface in water. The data has been processed by the graph-bootstrapping. I do not appreciate any major differences between Fig. 10 and the original data/findings by Söngen (Figs. 2 and 3). However, the fact that the authors have applied their method to AFM data from other groups is highly commendable. Based on all the revisions, I consider the Ms. suitable for publication in NCOMMS.

Reviewer #2 (Remarks to the Author):

This version includes data from Söngen et al. (PRL 120,116101 (2018)) that shows an atomically-resolved image of a calcite surface in water. The data has been processed by the graph-bootstrapping. I do not appreciate any major differences between Fig. 10 and the original data/findings by Söngen (Figs. 2 and 3). However, the fact that the authors have applied their method to AFM data from other groups is highly commendable. Based on all the revisions, I consider the Ms. suitable for publication in NCOMMS.

We thank the Reviewer for his/her recommendation.